

# An intrusion detection system based on convolution neural network

Yanmeng Mo, Huige Li, Dongsheng Wang and Gaqiong Liu

School of Computer, Jiangsu University of Science and Technology, Zhenjiang, China

## ABSTRACT

With the rapid extensive development of the Internet, users not only enjoy great convenience but also face numerous serious security problems. The increasing frequency of data breaches has made it clear that the network security situation is becoming increasingly urgent. In the realm of cybersecurity, intrusion detection plays a pivotal role in monitoring network attacks. However, the efficacy of existing solutions in detecting such intrusions remains suboptimal, perpetuating the security crisis. To address this challenge, we propose a sparse autoencoder-Bayesian optimization-convolutional neural network (SA-BO-CNN) system based on convolutional neural network (CNN). Firstly, to tackle the issue of data imbalance, we employ the SMOTE resampling function during system construction. Secondly, we enhance the system's feature extraction capabilities by incorporating SA. Finally, we leverage BO in conjunction with CNN to enhance system accuracy. Additionally, a multi-round iteration approach is adopted to further refine detection accuracy. Experimental findings demonstrate an impressive system accuracy of 98.36%. Comparative analyses underscore the superior detection rate of the SA-BO-CNN system.

## INTRODUCTION

In today's digital world, the exponential growth of data presents a significant challenge to network security. The ever-increasing complexity of network attacks has made it increasingly difficult to develop effective detection tools (*Sowmya & Anita, 2023*; *Ahmetoglu & Das, 2022*; *Friedberg et al., 2015*). Prominent incidents like the Triton malware attack, which crippled the security system of a petrochemical plant in Saudi Arabia, endangering lives, the Facebook information leak in 2018, and the SolarWinds supply chain attack in 2020, have demonstrated the immense economic losses suffered by enterprises and individuals, while highlighting the substantial threats to network security (*Alladi, Chamola & Zeadally, 2020*; *Kabir et al., 2018*). Consequently, traditional intrusion detection systems (IDS) are no longer sufficient in effectively combating these increasingly sophisticated network attacks. Urgently, there is a pressing need for a more intelligent and efficient security protection system worldwide.

The current popular intrusion detection method is to reduce the error rate by using different machine learning techniques. *Kabir et al. (2018)* proposed a novel intrusion

Corresponding author
Huige Li, 18605247081@163.com

detection method based on least squares support vector machine (LS-SVM) sampling. This approach consists of two stages: firstly, establishing an optimal allocation scheme, and then utilizing LS-SVM to detect the extracted samples (*Rathore et al., 2017*). In the area of network intrusion detection, *Htun & Khaing (2012)* employed random forest as the benchmark model and combined it with pattern recognition techniques to enhance the effectiveness of intrusion detection (*Tankard, 2011*). *Akhtar et al. (2023)* integrated data analysis technology with four robust machine learning ensemble algorithms, including the voting classifier, Bagging classifier, gradient boosting classifier, and the Bagging algorithm based on random forest, to create and test models using a network dataset. *Hidayat, Ali & Arshad (2023)* proposed a hybrid feature selection technique composed of the Pearson correlation coefficient and random forest model. For the machine learning model, decision tree, AdaBoost, and k-nearest neighbor were trained and tested on the TON_IoT dataset. The findings demonstrated the effectiveness of these machine learning techniques in detecting network intrusions (*Hidayat, Ali & Arshad, 2023*). *Turukmane & Devendiran (2024)* classified the different types of attacks by a hybrid machine learning model called Mud Ring assisted multilayer support vector machine (M-MultiSVM) and finally, the hyperparameters were tuned by the Mud Ring optimization algorithm. The proposed M-MultiSVM model could efficiently detect intrusion in the network. The performance metrics showed that the proposed system achieved 97.535% accuracy by using the UNSW-NB15 dataset (*Turukmane & Devendiran, 2024*). Traditional machine learning methods are effective in intrusion detection, but they also have limitations, because the traditional machine learning technology needs to artificially construct sample features. Its performance is dependent on its quality. In order to solve this problem, researchers have introduced deep learning techniques. Deep learning, a sub-field of machine learning, employs multi-layer neurons to model the learning process, thereby forming a more intricate artificial neural network (*Riyaz & Ganapathy, 2020*). Deep learning offers several advantages, such as automatic feature learning, even with large amounts of data (*Kocher & Kumar, 2021*). Consequently, compared with traditional machine learning, IDS based on deep learning typically demonstrate higher processing efficiency and detection accuracy. For instance, *Gao et al. (2014)* applied deep trust network in intrusion detection and achieved better results than other traditional machine learning methods. *Raman et al. (2017)* applied probabilistic neural networks to detection techniques. *Peddabachigari et al. (2007)* proposed a hybrid intrusion detection model based on deep learning and verified that the model is more efficient than traditional machine learning methods. Similarly, *Khan et al. (2023)* reported an intelligent IDS for IoT networks based on deep learning algorithms. The proposed model consisted of a recurrent neural network and gated recurrent units (RNN-GRU), which could classify attacks across the physical, network, and application layers. The results showed that the proposed system achieved an accuracy of 99% for network flow datasets and 98% for application layer datasets, superior to various traditional machine learning techniques (*Khan et al., 2023*). Bakhsh et al. created a deep learning-based IDS using feed forward neural networks (FFNN), long short-term memory (LSTM), and random neural networks (RandNN) to protect IoT networks from cyberattacks. the FFNN can handle complex IoT network traffic patterns, while the LSTM is good in capturing long-term

dependencies present in the network traffic. The proposed technique exhibits superior performance when compared with the traditional machine learning techniques; such as Naive Bayes, decision tree, random forest, and k nearest neighbor algorithms (*Bakhsh et al., 2023*). However, the detection rate of existing solutions is not very high, which also makes the security crisis still exist. The convolutional neural network (CNN) model is quite often utilized for solving research problems in fields like computer vision (*Gururaj, Vinod & Vijayakumar, 2023*; *Javanmardi et al., 2021*), image processing (*Hossain et al., 2023*; *Towfek & Khodadadi, 2023*), *etc.* due to its capability to extract location invariant features automatically. The application of CNN for IDS is not explored much.

Based on these above issues, to enhance resilience against intricate network attacks, we propose a system model named sparse autoencoder-Bayesian optimization-convolutional neural network (SA-BO-CNN), rooted in the CNN strategy. This article presents two main contributions: (1) A well-established data processing pipeline comprising one-hot encoding, resampling, and SA. SA effectively addresses the challenge of inadequate manual feature extraction, while resampling mitigates initial data imbalances; (2) Performance enhancement verified through experimental results, showcasing Bayesian optimization's significant impact on accuracy, precision, recall, and F1 score. This underscores the efficacy of Bayesian technology in bolstering IDS effectiveness, thereby fortifying network security. Our CNN-based IDS model, optimized with Bayesian techniques, exhibits superior adaptability across diverse datasets, achieving a remarkable worst-case accuracy rate of 98.36%.

# OVERVIEW OF INTRUSION DETECTION SYSTEM METHODS

## Convolution neural network

CNN, a deep learning model, is primarily utilized for processing and analyzing data structured in a grid format, such as images and videos. Drawing inspiration from the biological vision system, CNN autonomously learns and extracts features from images, rendering it a valuable asset in the realm of computer vision.

CNN comprises core components, including the convolution layer, pooling layer, activation function, and full connection layer. The convolution layer employs convolution operations on input data to discern features within the image, such as edges, textures, and shapes. Concurrently, the pooling layer reduces data dimensionality, decreases computational complexity, and extracts essential features. The Activation Function introduces nonlinearity, enabling the model to capture more intricate patterns and relationships. Lastly, the full connection layer maps the final feature map to the network's output layer for tasks such as classification or regression.

CNN boasts advantages such as parameter sharing, sparse connection, and a hierarchical structure, which significantly enhance its efficiency in image processing tasks (*Sharma et al., 2023*). Its ability to automatically learn image features sans manual extraction lends itself well to various computer vision tasks, including image classification, object detection, face recognition, and image generation. Furthermore, the application scope of CNN has expanded to encompass speech processing, natural language processing, and other

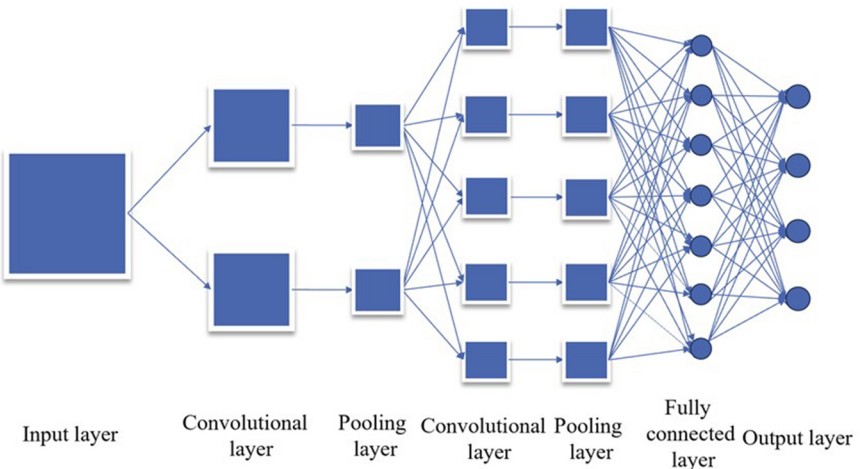

**Figure 1** **Convolution neural network structure diagram.**

domains, solidifying its status as a pivotal tool in the deep learning landscape. Figure 1 provides a visual depiction of its structure.

### Convolution layer

Convolutional layers constitute indispensable components in deep learning, primarily tailored for processing images, videos, and other two-dimensional data (*Yamashita et al., 2018*). They serve the critical function of extracting features from input data and find widespread application across diverse domains. Central to convolutional layers is the convolution operation, a fundamental technique that executes local weighted summation on input data, thereby capturing spatial structural information.

The distinctive advantage of convolutional layers lies in their efficacy at extracting features from input data through the utilization of local connections and parameter sharing. Local connections entail filters convolving solely with local regions of the input data, as opposed to the entire input. Meanwhile, parameter sharing dictates that each filter's parameters are shared across different locations in the input, employing the same set of weights. These strategies contribute to reducing the model's storage requirements while bolstering its generalization capability.

Moreover, convolutional layers have the capacity to generate multiple feature maps by employing multiple filters, with each feature map corresponding to a specific filter. This multi-feature map approach enables the capture of variations across different feature directions within the input data, thereby facilitating the extraction of richer feature representations. By stacking multiple convolutional layers, the network progressively extracts higher-level abstract features, facilitating tasks such as classification, regression, and more in subsequent fully connected layers (FCLs) or output layers.

In essence, convolutional layers play an indispensable role in deep learning by efficiently extracting features from input data and exerting a pivotal influence across various domains. Through the mechanisms of local connections, parameter sharing, and multi-channel

outputs, convolutional layers empower deep learning models to capture spatial structural information from input data and furnish rich feature representations.

The formula of convolution layer is as follows:

$$y_{i,j,k} = \sum_{u=0}^{k-1}\sum_{v=0}^{k-1}\sum_{c=0}^{C-1} w_{u,v,c,k} x_{(i+u),(j+v),c}. \tag{1}$$

In this case, $y_{i,j,k}$ represents the convolutional output of the $k-th$ kernel at position $(i,j)$, $w_{u,v,c,k}$ represents the weight at position $(u,v)$ of the $c-th$ channel of the $k-th$ kernel, and $x_{(i+u),(j+v),c}$ represents the value of the $c-th$ channel at position $(i,j)$ in the input data.

### Pool layer

Pooling layers are integral components of CNNs and hold significant academic value in image processing and computer vision domains (*Nasr-Esfahani et al., 2019*). Their primary function is to reduce the spatial dimensions of feature maps through downsampling operations, thereby diminishing the number of parameters and computational complexity. This dimensionality reduction not only enhances computational efficiency but also facilitates the extraction of primary features from the input data.

Within pooling layers, the most prevalent operations are max pooling and average pooling. Max pooling retains the most significant features of input regions by selecting the maximum value within each subregion, while average pooling calculates the average value of features within each subregion, thereby blurring details and mitigating the influence of noise. Both of these pooling operations contribute to extracting crucial information from the input data, providing more robust and abstract feature representations for subsequent layers. This article primarily adopts the max pooling method.

Furthermore, pooling layers exhibit translation invariance, meaning that irrespective of the features' positions in the input data, the pooling operation can still detect their presence and effectively aggregate them. This characteristic renders pooling layers resilient to translation, rotation, and scale changes in images, thereby enhancing the model's generalization ability (*Mumuni & Mumuni, 2021*; *Zhang et al., 2019*).

In summary, pooling layers play an essential role in CNNs, offering potent feature extraction and abstraction capabilities for image processing tasks by reducing spatial dimensions, extracting primary features, and providing translation invariance. In both academic research and practical applications, the design and optimization of pooling layers are crucial for enhancing model performance and efficiency. The formula for the pooling layer is as follows:

$$y_{i,j,k} = max_{u=0}^{p-1} max_{v=0}^{p-1} x_{(i\times s+u),(j\times s+v),k} \tag{2}$$

where $y_{i,j,k}$ represents the output of the $k-th$ channel after pooling, $s$ represents the step size, and $p$ represents the size of the pooled area.

### Fully connected layer

FCLs are pivotal elements within deep neural networks, facilitating highly flexible and expressive feature extraction and transformation (*Li et al., 2023*; *Matsumura et al., 2023*).

They achieve this by establishing connections between every neuron from the preceding layer to each neuron in the subsequent layer, with each connection possessing an associated weight that governs the impact of the previous layer's output on the current layer's neurons.

FCLs exhibit numerous merits. Firstly, they boast potent expressive capabilities, enabling them to discern complex patterns and capture nonlinear relationships within the input data, thereby extracting essential abstract features. Secondly, FCLs offer remarkable flexibility, allowing them to adapt to diverse complex tasks and data distributions, thereby affording a considerable degree of model freedom. Additionally, FCL computations are straightforward, facile to implement, and amenable to acceleration through efficient matrix operations.

The roles of FCLs can be succinctly delineated as follows: initially, they transmute the original input data into higher-level abstract features by learning weights and biases, thereby extracting critical information. Subsequently, FCLs are commonly deployed in the final layer of neural networks to map the extracted features to specific classes or value ranges, thereby facilitating classification and regression tasks. Finally, FCLs undertake comprehensive evaluations on the input data and formulate decisions and predictions based on the acquired weights.

The formula for the full connection layer is as follows:

$$y = f(Wx + b) \tag{3}$$

where $y$ is the output vector, $x$ is the input vector, $W$ is the weight matrix, $b$ is the offset vector, and $f$ is the activation function.

## Data preparation

The NSL-KDD dataset stands as a well-known and extensively utilized resource for network intrusion detection, serving researchers with a realistic and diverse network traffic dataset to evaluate various network IDS (*Gurung, Ghose & Subedi, 2019*; *Choudhary & Kesswani, 2020*). It represents an enhanced and extended iteration of the original KDD Cup 1999 dataset, featuring additional attack types and streamlined data, thereby enriching its complexity. Comprising authentic network traffic data, the NSL-KDD dataset encompasses diverse attack categories such as DoS, R2L, U2R, and Probing, rendering it more representative and pragmatic.

To bolster the dataset's quality and usability, several preprocessing steps have been meticulously executed. These include the removal of duplicate data, standardization of data formats, and elimination of redundant features, effectively addressing dataset imperfections and rendering it more conducive for network intrusion detection research and application. Moreover, the dataset is conventionally partitioned into training and testing sets, facilitating the development and evaluation of novel network intrusion detection techniques while assessing algorithmic performance and generalization capabilities.

The widespread utilization of the NSL-KDD dataset in researching and evaluating network intrusion detection algorithms underscores its significance. Researchers leverage this dataset to gauge the effectiveness of various network IDS and explore avenues for enhancing network security and defense capabilities. Furthermore, the NSL-KDD dataset

**Table 1 Comparison of KDD Cup 99 and NSL-KDD data sets.**

| Dataset name | KDD Cup 99 | NSL-KDD |
|---|---|---|
| Number of training sets (articles) | 4,898,431 | 125,973 |
| Number of test sets (pieces) | 311,029 | 22,544 |
| Number of network attack features | 41 | 42 |
| Is the data redundant | Yes | No |
| Dataset time | 1998 | 1999 |

serves as a valuable resource and reference for real-world network security concerns, contributing substantially to the advancement and implementation of network security technologies.

Table 1 offers a comparative analysis between the KDD Cup 99 dataset and the NSL-KDD dataset.

## Sparse autoencoders

Due to the dataset's extensive feature set, we employ feature dimensionality reduction methods to enhance computational efficiency. One such method is the autoencoder, an unsupervised learning algorithm utilized for both data dimensionality reduction and feature extraction. Comprising an encoder and a decoder, it compresses input data into a low-dimensional representation while faithfully reproducing the original data in its output (*Liu et al., 2022*).

A variant of the autoencoder, the Sparse Autoencoder, aims to produce sparse encoding representations while acquiring effective features. This variant typically generates encoding vectors with only a small number of non-zero elements, with the remaining elements approximating zero. Through the incorporation of a sparsity constraint, sparse autoencoders excel in discerning meaningful features and adeptly managing noise and redundant information within input datasets.

$$\sum_{j=1}^{l} p\log(p/\mu_j) + (1-p)\log[(1-p)/(1-\mu_j)] \tag{4}$$

where $l$ represents the number of neurons in the hidden layer, "$\mu_j$" represents the average output value of the $j$-th neuron in the hidden layer, which is also known as the average activation. "$p$" is a small value. Here, the penalty factor is a relative entropy between two Bernoulli random variables, "$p$" and "$\mu_j$". Relative entropy is a measure of the difference between two distributions. When the two distributions are equal, the relative entropy is zero. As the difference between the distribution's increases, the relative entropy also increases. Therefore, during the optimization process of the loss function with the penalty factor, the average activation of the neurons in the hidden layer of the sparse autoencoder will approach the value of "$p$". In this study, a sparse autoencoder is primarily used for data dimensionality reduction and feature extraction.

## Bayesian optimization

Bayesian optimization, an optimization technique rooted in Bayesian statistics, constructs a posterior probability model of the objective function to guide the selection of

candidate points for evaluation through sampling, thus progressively seeking the optimal solution (*Guo et al., 2023*). Essentially treating the objective function as a black box, with hyper-parameters (*e.g.*, learning rate and regularization coefficient in machine learning algorithms) as input and an index (*e.g.*, cross-validation effect) as output, Bayesian optimization iteratively updates the posterior probability distribution using historical data. It then selects new hyper-parameter combinations for evaluation based on this distribution and integrates them into the historical dataset, gradually converging towards the global optimal solution (*Qiao et al., 2023*).

Distinguished by its ability to yield superior solutions within a relatively small number of iterations and tackle complex optimization problems such as high-dimensional and non-convex scenarios, Bayesian optimization stands out among other optimization methods. Consequently, it finds widespread application across various domains, including hyper-parameter tuning and automated machine learning. The process of Bayesian parameter optimization typically involves three key steps:

(1) Choose a priori function to express the hypothesis about the optimized function. The Gaussian process used in this article is a set of random variables, and any finite random variables satisfy a joint Gaussian distribution. If X represents the training set $\{x_1, x_2, \ldots\ldots, x_t\}$, f represents the unknown function value set $\{f(x_1), f(x_2), \ldots, f(x_t)\}$, $\Sigma$ represents the covariance matrix $\prod$ formed by $k(x, x\prime)$ and $\theta$ represents the hyperparameter, when there is observation noise and assumes that the noise $\varepsilon$ satisfies the independent and identically distributed Gaussian distribution $p(\varepsilon) = N(0, \sigma^2)$, the marginal likelihood distribution can be obtained as follows:

$$P(y|X, \theta)_2 = \int p(y|f)p(f|X, \theta)df = N(0, \Sigma + \sigma^2 I) \tag{5}$$

where $y$ represents the collection of observations $\{y_1, y_2 \ldots\ldots y_t\}$ .

(2) By maximizing the marginal likelihood distribution by ML maximum likelihood estimation, the optimal hyperparameter is obtained, and the prior distribution $p(\theta)$ is given to the hyperparameter. According to Bayesian theorem, the following results are obtained:

$$P(\theta|D_{1:t}) = \frac{p(D_{1:t}|\theta)P(\theta)}{p(D_{1:t})} \tag{6}$$

(3) According $\hat{\theta}_t$ to the specific acquisition function that can be obtained:

$$\hat{\alpha}_t(x) = \alpha(x; \theta_t). \tag{7}$$

# INTRUSION DETECTION SYSTEM BASED ON CONVOLUTION NEURAL NETWORK

The article primarily utilizes CNN for prediction and intrusion detection on the NSL-KDD dataset, chosen for its suitability in analyzing network connection features, handling high-dimensional data, facilitating pattern recognition and feature extraction, and demonstrating robustness and generalization capabilities.

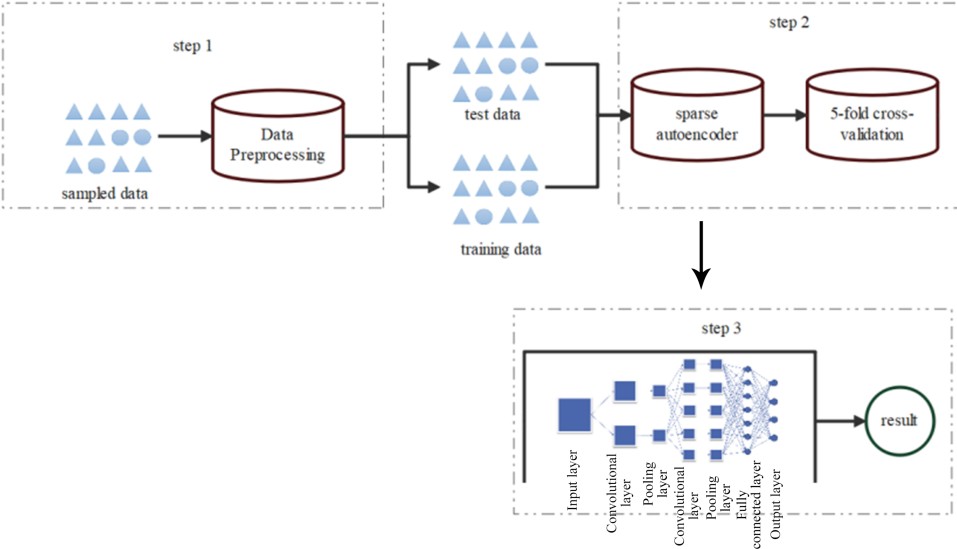

**Figure 2** **Framework diagram of CNN model construction.**

By integrating convolutional layers and pooling layers, CNN effectively captures spatial structures and local relationships within the NSL-KDD dataset, autonomously learning and extracting crucial features that enable efficient classification and discrimination of various attack types. Moreover, CNN reduces the number of model parameters, thus decreasing computational complexity and showcasing resilience to slight variations in input data, making it an ideal choice for addressing challenging prediction tasks in the NSL-KDD dataset.

The model building framework diagram is depicted in Fig. 2.

The detection process mainly involves three steps:

In the first step, symbolic features of the sampled dataset are numerically encoded, followed by SMOTE resampling of the preprocessed data. Subsequently, all preprocessed data is normalized to obtain standardized raw data.

In the second step, a sparse autoencoder feature extraction model is applied to preprocess high-dimensional and nonlinear data, aiming to reduce original data dimensionality while preserving optimal representation.

In the third step, the optimal low-dimensional representation of the original data serves as input for the classifier, which identifies normal network data and various types of network attack data.

The key innovation of our SA-BO-CNN model lies in combining the Bayesian optimization algorithm with self-supervised learning and utilizing sparse autocoded data for model training.

**Table 2  Protocol_type one-hot coding.**

| Characteristic of belonging character type | Category | After one-hot coding | | |
|---|---|---|---|---|
| | TCP | 1 | 0 | 0 |
| protocol_type | UDP | 0 | 1 | 0 |
| | ICMP | 0 | 0 | 1 |

## Data preprocessing
### One-hot coding

The NSL-KDD dataset comprises both numerical and categorical variables, yet deep learning algorithms inherently handle only numerical data, necessitating specialized techniques for managing categorical variables. Here, we delve into the process of independent one-hot encoding, exemplified by the character feature "protocol_type" (protocol type):

One-hot encoding is a method that converts categorical variables into digital formats amenable to machine learning algorithms. Essentially, it treats each possible value as a distinct binary feature. For a given data point, the feature corresponding to its value is assigned a value of 1, while features for other values are set to 0. The objective is to transform original text-based categorical features into binary sparse vectors, enabling deep learning algorithms to interpret and utilize them effectively.

In this context, there exist three distinct protocol types: "tcp", "udp", and "icmp." After one-hot encoding, the "protocol_type" feature undergoes transformation into a sparse vector with a dimension equal to the number of protocol types, which is 3. This vector represents the protocol type utilized in each example, as illustrated in Table 2. Each column corresponds to a specific protocol type, such as "tcp", "udp", or "icmp". If a particular protocol is employed in an example, the corresponding column is marked as 1; otherwise, it is designated as 0. This encoding methodology enables deep learning models to comprehend and process categorical variables, thereby facilitating subsequent analysis and modeling endeavors.

Upon examination of Table 2, it becomes evident that the character features encompass numerous categories. Each category undergoes independent one-hot encoding, yielding a corresponding sparse vector. For example, the "service" feature encompasses 70 distinct categories, leading to the generation of 70-dimensional sparse vectors post one-hot encoding. Similarly, the "flag" feature comprises 11 different categories, resulting in 11-dimensional sparse vectors post one-hot encoding. Consequently, character-type discrete features can be expanded to a certain extent through one-hot encoding.

Furthermore, the NSL-KDD dataset encompasses various attack features that can be broadly categorized into four primary categories, each with its own subcategories. A detailed breakdown of these categories is presented in Table 3.

It can be seen from the table that there are 39 sub-attack type tags in NSL-KDD data set tags. In order to prevent dimension explosion caused by single hot coding, each sub-class is now divided into five categories, namely DOS, Probe, U2R, R2L and Normal. The specific division rules are shown in Table 3 above.

**Table 3  Attack characteristic division.**

| Attack type | Attack subtype | Total |
| --- | --- | --- |
| DOS | back, land, neptune, pod, smurf, teardrop,apache2, mailbomb, processtable, udpstorm,worm | 11 |
| Probe | ipsweep, nmap, portsweep, satan, saint, mscan | 6 |
| U2R | buffer_overflow, loadmodule, perl, rootkit, sqlattack, xterm,ps | 7 |
| R2L | ftp_write, guess_passwd,httptunnel, imap, multihop, named,phf, spy, sendmail, snmpgetattack,warezclient, warezmaster, snmpguess,xlock,xsnoop | 15 |

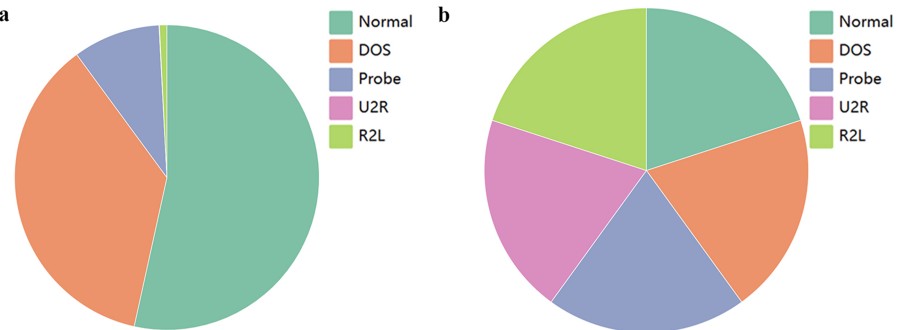

**Figure 3  Data distribution.** (A) Initial data; (B) resampling data.

### SMOTE resampling

In deep learning, the issue of imbalanced data can result in inadequate predictive performance of the model for certain categories. In the NSL-KDD dataset, some attack categories have a limited number of samples, leading to insufficient attention from the model during training. Even when the model is trained on a large-scale dataset, the influence of dominant samples may introduce bias towards the categories they represent. The specific scenario is illustrated in Fig. 3A.

Figure 3A and Table 4 show that the number of DOS attack samples (45927) is significantly higher than that of the other three attack types (52 −11656), with U2R attack having the lowest number of samples (52), making it difficult to discern even with magnification.

To combat the issue of imbalanced data in machine learning tasks, the Synthetic Minority Over-sampling Technique (SMOTE) is an effective data balancing technique. Imbalanced data refers to a scenario where there is a significant difference in the number of samples between different classes in the training set, with one class having far more samples than the other classes. In such cases, models tend to favor the majority class, resulting in insufficient recognition of the minority class.

SMOTE addresses this problem by generating synthetic samples of the minority class to balance the dataset. It interpolates the minority class samples and generates artificial samples that are similar but not identical to the original samples. These synthetic samples

**Table 4  Distribution of attack types in the dataset before and after SMOTE resampling.**

| Class | Count | |
|---|---|---|
| | Before SMOTE resampling | After SMOTE resampling |
| Normal | 67,343 | 67,342 |
| U2R | 52 | 67,342 |
| Probe | 11,656 | 67,342 |
| DoS | 45,927 | 67,342 |
| R2L | 995 | 67,342 |

fill the gaps in the original data, improving the balance of the dataset by increasing the number of minority class samples.

Resampling data with SMOTE offers several benefits. Firstly, it can enhance model performance by improving the classifier's ability to recognize the minority class. By increasing the number of minority class samples, the model can better learn the features and patterns of the minority class during training, reducing misclassifications and improving overall classifier performance.

Secondly, SMOTE resampling can mitigate classifier bias towards the majority class. Due to the excessive number of majority class samples, models tend to predict the majority class more frequently, overlooking important information from the minority class. By generating synthetic samples, SMOTE balances the sample sizes between different classes, reducing this bias and enabling the classifier to treat different classes more equally.

Furthermore, the synthetic samples produced by SMOTE retain the characteristics of the original samples while introducing a certain degree of diversity. This increases the diversity of the data, provides more training samples, and enhances the model's generalization ability (*Shrinidhi, Kaushik Jegannathan & Jeya, 2023*).

The fundamental principle of SMOTE revolves around gaining a deep understanding of a few categories of samples and subsequently generating new samples based on these selected ones to augment the dataset. The specific steps involved in this operation are as follows: (1) Selection: Randomly choose a sample point, denoted as X, from the selected categories. (2) Neighbor calculation: Determine the k nearest neighbors of the sample point in the feature space using primarily the Euclidean distance algorithm. (3) Generation of new samples: For each neighbor, create one or more composite samples based on their dissimilarities. The attributes of the new sample are obtained through linear interpolation between the attributes of the original sample and those of the neighbor sample. (4) Repetition: Repeat the above process until a sufficient number of synthetic samples have been generated.

The primary objective of the SMOTE algorithm is to enhance the performance and accuracy of the model by rebalancing the class distribution within the dataset through the iterative execution of the aforementioned steps. The key functions of the SMOTE algorithm are as follows:

$$X_{new} = x + rand(0, 1) * |x - xn|. \tag{8}$$

Figure 3B demonstrates the outcome after applying SMOTE resampling, and it appears that the sample quantities among different classes have achieved a relatively balanced

distribution. It can be seen that an equilibrium state has been reached between the classes, and the number of samples has increased from 125,973 to 336,710.

Finally, after resampling the data, the data is normalized. Normalization of features is to scale the range of different features to the same scale or range in order to eliminate the deviation or adverse effects caused by the difference of eigenvalues. In this article, min-max normalization is adopted, and its specific formula is as follows

$$y_{min-max} = \frac{x - min(x)}{max(x) - min(x)}.$$ (9)

The normalized results are all positive numbers and the data is scaled or mapped to the interval (0,1). The final results are stored in Excel table to facilitate subsequent operation and analysis. This series of steps helps to improve model performance, especially when dealing with uneven data.

## Construction of intrusion detection system model

In this article, we employ a deep learning model evaluation technique known as "5% cross-validation" (*Yu et al., 2014*) to assess the performance of the SA-BO-CNN model and determine suitable parameters for classification training. This method involves processing Excel table data and adjusting the input data format to match the SA-BO-CNN model's requirements. Additionally, we introduce an objective function named "cnn_evaluate" to gauge the SA-BO-CNN model's performance and derive the average accuracy from cross-validation.

During the five-fold cross-validation process, we randomly partition the NSL-KDD dataset into five segments, utilizing four segments for training and one for validation in each iteration. Subsequently, we train the model with the training data and evaluate its performance on the validation set, measuring metrics such as accuracy and precision. This procedure iterates five times to ensure each segment serves as the validation set, and the results from these validations are averaged to derive the final model evaluation metric.

This approach offers several advantages, including comprehensive data utilization, mitigating bias in model evaluation, and precise estimation of model performance. Consequently, it aids researchers in accurately selecting deep learning model parameters, thereby enhancing model performance and generalization capability.

In this study, we construct a model comprising two convolutional layers, employing ReLU as the activation function for both convolutional and fully connected layers. ReLU, widely utilized in deep learning, excels in capturing complex nonlinear relationships within data, providing efficient representation, reducing redundancy in computations and parameters, and addressing overfitting concerns. Despite drawbacks such as dead neurons and non-centered output, ReLU remains popular in deep learning tasks due to its effectiveness and practical utility.

For the output layer, given our multi-classification problem, we utilize the Softmax activation function. Softmax offers advantages such as transforming raw outputs into a probability distribution, handling multi-class scenarios by normalizing outputs across classes, complementing cross-entropy loss, and simplifying gradient computations (*Banerjee et al., 2020*; *Zhu et al., 2020*). Additionally, we incorporate L2

regularization to manage model complexity, mitigate overfitting risks, and enhance model performance and generalization by gradually reducing less significant feature weights and sometimes setting them to zero, facilitating feature selection and interpretability while limiting parameter magnitudes and enhancing model stability.

Furthermore, we define a hyperparameter search space termed "pbounds", encompassing ranges for five parameters: convolutional kernel count and size, pooling size, fully connected layer node count, and learning rate. Subsequently, we instantiate a "Bayesian Optimization" object, passing the objective function and search space, to identify optimal hyperparameters through the Bayesian optimization process. This approach aids in determining the most suitable hyperparameter configuration, thereby bolstering model performance and generalization.

## Experiments and results
### Parameter tuning

The experimental environment configuration is shown in Table 5.

The SA-BO-CNN model improves its performance and generalization ability by adjusting various parameters, including hyperparameters, network structure, and data augmentation. The goal of parameter adjustment is to enhance accuracy and effectiveness in specific tasks, prevent overfitting, and optimize the training process and generalization capability of the model. It is an iterative process that requires repeated experiments and adjustments based on the actual situation to find the best configuration.

There are two commonly used methods for parameter tuning: grid search and Bayesian optimization. Grid search exhaustively searches through a finite set of parameter combinations, but it can be time-consuming and prone to the curse of dimensionality. Therefore, this article primarily utilizes the Bayesian optimization method to adjust parameters such as the number and size of convolution kernels, pooling size, number of nodes in the fully connected layer, and learning rate.

The first parameter being tuned is the learning rate. If the learning rate is too high, the model's parameters may oscillate or diverge along the gradient direction, whereas if it is too low, the training convergence will be slow and may even get trapped in local optima. In the tuning process, the learning rate is adjusted within the range of (0.001, 0.01), and cross-validation is employed to evaluate the model's performance under different learning rates. The debugging process is illustrated in Fig. 4.

The second parameter being tuned is the size of the convolution kernel. In this article, the adjustment range is set to (3, 5). The size of the convolution kernel affects the degree of abstraction of features and the receptive field size of the neural network. If the selected convolution kernel is too small, the neural network may not effectively capture large-scale features in the image. Conversely, if the selected convolution kernel is too large, the neural network may become too sensitive to detail information, which can lead to overfitting problems. The debugging process is illustrated in Fig. 5.

The third parameter being tuned is the number of convolution kernels. The feature extraction ability of the neural network can be adjusted by changing the number of convolution kernels. A lower number of convolution kernels may limit the neural network's

| Table 5 | Experimental environment. |
| --- | --- |
| **Name** | **Configuration information** |
| Operating system | Win 11 |
| Development language | Python 3.9.13 |
| Framework | TensorFlow + Keras |
| CPU | 12th Gen Intel(R) Core(TM) i7-12700H |
| GPU | NVIDIA GeForce RTX 3050 Ti Laptop GPU |

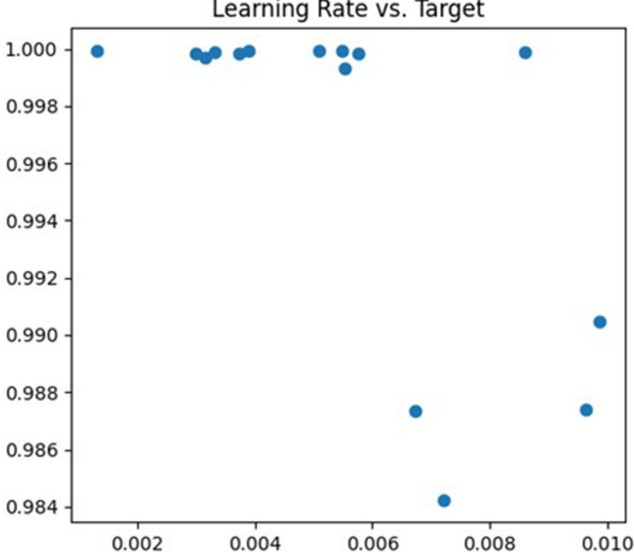

**Figure 4**   **Learning rate debugging process.**

ability to learn complex features, resulting in underfitting. On the other hand, a higher number of convolution kernels may increase the neural network's expressive power, but it also increases the risk of overfitting. The debugging process is illustrated in Fig. 6.

The fourth parameter being tuned is the pooling window size of the Pooling Layer. Selecting a pooled window size that is too large can result in losing critical information and important features. On the other hand, if it is too small, the model may become susceptible to noise or overfitting. In this article, the adjustment range is set to (2, 4). The debugging process is illustrated in Fig. 7.

The fifth parameter being tuned is the number of nodes in the fully connected layer. The fully connected layer is typically used to flatten the output features from the preceding convolution or pooling layers and perform tasks like classification or regression. Modifying
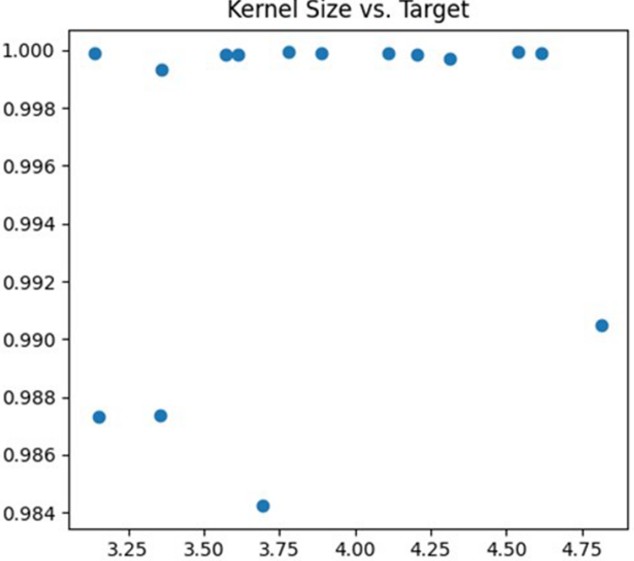

**Figure 5** **Kernel size debugging process.**

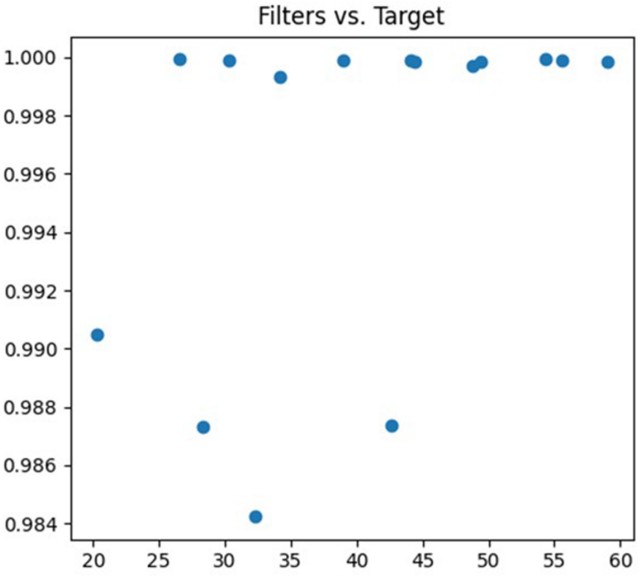

**Figure 6** **Filters debugging process.**

the number of nodes in the fully connected layer can influence the model's complexity. The debugging process is depicted in Fig. 8.

The change of accuracy and loss rate of the model after 10 iterations of Bayesian optimization is shown in Fig. 9.

The model tuning results are presented in Table 6. Based on the table, it is evident that this CNN-based model achieves an impressive accuracy of 98.36%. This accomplishment
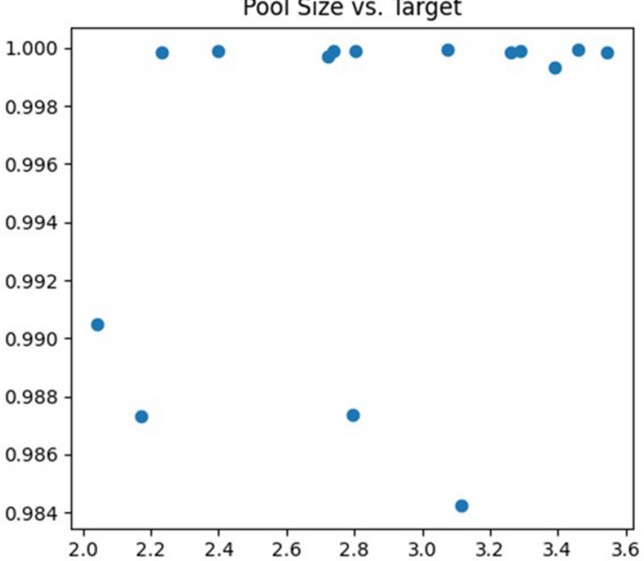

**Figure 7    Pool size debugging process.**

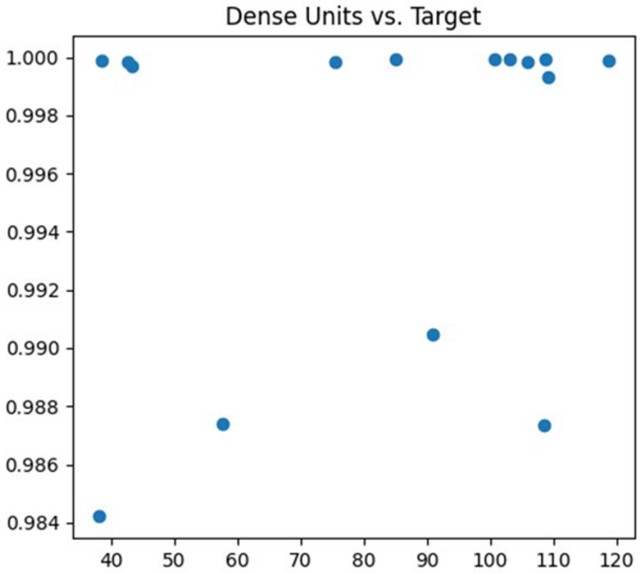

**Figure 8    Dense units debugging process.**

is attributed to the utilization of a 3×3 convolutional kernel with a count of 40, a 3×3 convolution kernel with a count of 61, pooling windows of size 3×3 and 3×3, a learning rate of 0.000553, along with fully connected layer nodes of 41, respectively.

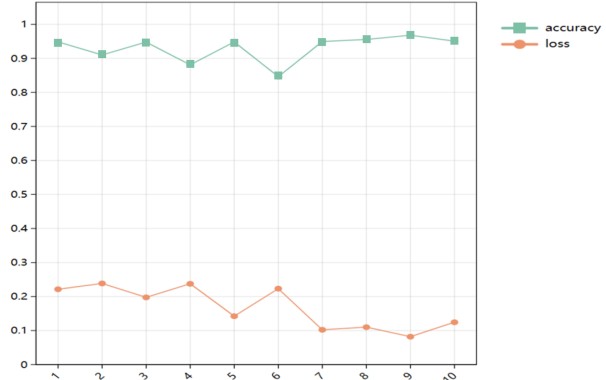

**Figure 9** Changes of accuracy and loss rate.

**Table 6** Results of optimal parameters.

| Accuracy | Convolution kernel size | Filters | Pooled window size | Learning rate | Number of nodes in full connection layer |
|---|---|---|---|---|---|
| 98.36% | 3×3 | 40 | 3×3 | 0.000553 | 41 |
| / | 3×3 | 61 | 3×3 | / | / |

### Model validation and analysis

In this article, we mainly use Precision, Recall and F1-score to evaluate the performance of the model.

For a multi-classification problem, it is often necessary to number the categories, have K different categories, and express the i-th category as $C_i$. At the same time, suppose there are $N_j$ samples with the true class label of $C_j$, among which $j \in [1, k]$. The classifier assigns $M_i$ samples to class $C_i$, of which, $i \in [1, k]$.

Then the formulas of accuracy rate, recall rate and F1 score under multi-classification problem are as follows:

Precision rate:

$$Precision = \frac{M_i}{\sum_{i=1}^{k} M_i}. \tag{10}$$

Accuracy rate measures the proportion of the samples correctly classified by the classifier in all the samples classified into this category.

Recall rate (Recall):

$$Recall = \frac{M_i}{N_j}. \tag{11}$$

The recall rate measures the proportion of samples that the classifier can correctly classify in the real category.

F1 score:

$$F1 = \frac{2 \times Precision \times Recall}{Precision + Recall}. \tag{12}$$

**Table 7 Evaluation effect under different algorithms.**

| Algorithm name | Accuracy | F1 score | Precision rate | Recall rate |
|---|---|---|---|---|
| SA-BO-CNN | 0.9836 | 0.9900 | 0.9906 | 0.9900 |
| CNN-BiLSTM (*Shrinidhi, Kaushik Jegannathan & Jeya, 2023*) | 0.8358 | 0.8114 | 0.7965 | 0.8039 |
| Naive Bayes (*Yu et al., 2014*) | 0.8885 | 0.906 | 0.913 | 0.899 |
| CNN-IDMDI (*Banerjee et al., 2020*) | 0.9873 | 0.9874 | 0.9875 | 0.9873 |

F1 score is a comprehensive index of precision rate and recall rate, which weights and averages precision rate and recall rate, and obtains the maximum value when precision rate and recall rate are equal.

Under the condition of optimal parameters, the corresponding evaluation results of this article are shown in Table 7.

Based on Table 7, our SA-BO-CNN model demonstrates superior performance compared to other models. The model reported by *Jiang et al. (2020)* lacks a parameter tuning function, leading to an over-reliance on parameter selection and subsequently yielding a low precision rate. In addition, the study by *Oluwakemi, Muhammad & Anyachebelu (2023)* has undertaken more comprehensive efforts. They evaluate the efficacy of three distinct machine learning algorithms—CNN, recurrent neural networks (RNN), and Naive Bayes—in identifying diverse attack categories. The result indicates that CNN and RNN slightly outperform the naive Bayesian algorithms. The moderate performance is attributed to inherent limitations such as sensitivity to data noise and inability to process missing data. In *Gan et al. (2022)*, a method combining a gradient coordination mechanism and focus loss is proposed exhibiting high accuracy. Nevertheless, it demands extensive parameter adjustments, posing a challenge in parameter tuning. Therefore, achieving improved results entails conducting numerous experiments to ascertain the optimal parameter configuration.

## CONCLUSION

This article introduces a CNN-based intrusion detection system model, which aims to address the growing significance of network security issues. The model framework consists of three main components: feature selection, feature transformation, and classifier. To effectively handle features, various techniques are employed, such as one-hot coding, SMOTE resampling, and feature normalization. Through experimental results on the NSL-KDD sample set, the proposed CNN model demonstrates exceptional performance in detecting intrusions.

Notably, this CNN model exhibits high accuracy in monitoring different types of intrusions in network traffic. It particularly excels in handling the challenges of uneven and multi-type problems encountered in network intrusion monitoring. This outstanding capability positions it with great potential for practical applications in the field of network security. This deep learning model is expected to effectively combat network threats, enhance network security, reduce false positives, and play a pivotal role in today's complex

network environment. Future endeavors will center on crafting algorithms capable of not only detecting attacks but also implementing suitable protective measures, minimizing code runtime, enhancing transparency, aiding cybersecurity professionals in comprehending and addressing emerging threats, and effectively deploying them in real-world scenarios.

### Funding
This work was supported by the National Natural Science Foundation of China (Nos. 62072127, 62002076, and 61702234), and the Open Fund for Innovative Research on Ship Overall Performance (No. 25422217). The funders had no role in study design, data collection and analysis, decision to publish, or preparation of the manuscript.

### Grant Disclosures
The following grant information was disclosed by the authors:
The National Natural Science Foundation of China: 62072127, 62002076, 61702234.
The Open Fund for Innovative Research on Ship Overall Performance: No. 25422217.

### Competing Interests
The authors declare there are no competing interests.

### Author Contributions
- Yanmeng Mo conceived and designed the experiments, performed the experiments, analyzed the data, performed the computation work, prepared figures and/or tables, authored or reviewed drafts of the article, and approved the final draft.
- Huige Li conceived and designed the experiments, analyzed the data, performed the computation work, prepared figures and/or tables, authored or reviewed drafts of the article, and approved the final draft.
- Dongsheng Wang performed the experiments, prepared figures and/or tables, and approved the final draft.
- Gaqiong Liu performed the experiments, prepared figures and/or tables, and approved the final draft.

### Data Availability
The raw data and code are available in the Supplemental Files.

### Supplemental Information
Supplemental information for this article can be found online at http://dx.doi.org/10.7717/peerj-cs.2152#supplemental-information.

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
