# Peer review of "An intrusion detection system based on convolution neural network"

_PeerJ Computer Science, doi:10.7717/peerj-cs.2152_

## Round 0.1 · original submission · Major Revisions

The review process is now complete. While finding your paper interesting, the referees and I feel that more work could be done before the paper is published. My decision is therefore to provisionally accept your paper subject to major revisions. Note that this does not guarantee that your paper will be accepted after a revision. More details are needed.

**Language Note:** PeerJ staff have identified that the English language needs to be improved. When you prepare your next revision, please either (i) have a colleague who is proficient in English and familiar with the subject matter review your manuscript, or (ii) contact a professional editing service to review your manuscript. PeerJ can provide language editing services - you can contact us at [email protected] for pricing (be sure to provide your manuscript number and title). – PeerJ Staff

Reviewer 1 ·

Basic reporting

no comment

Experimental design

no comment

Validity of the findings

no comment

Annotated reviews are not available for download in order to protect the identity of reviewers who chose to remain anonymous.
Cite this review as

Reviewer 2 ·

Basic reporting

Authors have proposed a new model for CNN-based intrusion detection in this study. The literature review has been adequately presented, and the model details have been provided comprehensively. However, there are some concerns as follows:

- There is a semantic inconsistency in the sentence: "Deep learning offers several advantages, such as automatic feature learning, enabling the automatic extraction of features." Here, 'automatic feature learning' and 'automatic extraction of features' convey the same meaning, leading to redundancy. Similar writing errors are found in other parts of the manuscript, indicating the need for language editing for consistency and clarity.
- Although the literature review is up-to-date, the number of scanned articles is very limited. In order to better evaluate the position of the current study in the literature, the number of studies included in the literature review should be increased.
- The authors have not addressed the contributions of their study to the literature. What are the contributions of this study to the literature? These contributions should be listed in detail and thoroughly explained.

Experimental design

- The proposed CNN model is not described in detail. The sequence of convolution layers and where pooling layers are used remain unclear. It would be beneficial to provide verbal explanations in addition to the visual representation in Figure 2. Furthermore, the proposed CNN model seems to be overly simplistic. What is the innovative aspect of the proposed model? Is it its dual-branch structure, for instance? The design of the CNN model should also reflect the innovative aspect of the study.

Validity of the findings

- There are some nonsensical numbers related to the optimal parameters presented in Table 4. How can a kernel size be 4.2 in a CNN network? How can a pooled window size be 3.293? These numbers should be integers, not fractions. Another example is the number of convolution kernels used, how can it be 39.21? Additionally, some values are left blank in Table 4. Why are they empty? A table or figure should not only be presented but also explained in detail within the text.
- The proposed method has been compared with previous studies in the literature. Study [29] is from 2020, while study [30] is partly a survey study (including older studies within itself). In order to effectively evaluate the contribution to the literature, it would be beneficial to add recent studies to Table 5. Additionally, while the accuracy metric has been added to the abstract, it has not been included in the comparison table. Since accuracy is one of the most important metrics, it should be included in Table 5.

Additional comments

- It would be beneficial to include information about future plans in the conclusion section, such as what will be done in future studies. It is strongly recommended to add this information.
- The authors have only included the training phase as a code file. It would be beneficial to also include the test file and the saved model for the results obtained under different metrics to be visible for the proposed method

Cite this review as

---

## Round 0.2 · Minor Revisions

The review process is now complete. While finding your paper interesting and worthy of publication, the referees and I feel that more work could be done before the paper is published. My decision is therefore to provisionally accept your paper subject to minor revisions.

Reviewer 1 ·

Basic reporting

No comment.

Experimental design

No comment.

Validity of the findings

No comment.

Additional comments

No comment.

Annotated reviews are not available for download in order to protect the identity of reviewers who chose to remain anonymous.
Cite this review as

Reviewer 2 ·

Basic reporting

The authors wrote the following paragraphs in the revision. However, they say exactly the same thing. One of these needs to be removed.

" [1] By integrating convolutional layers and pooling layers, CNN effectively captures spatial structures and local relationships within the NSL-KDD dataset, autonomously learning and extracting crucial features that enable efficient classification and discrimination of various attack types. Moreover, CNN reduces the number of model parameters, thus decreasing computational complexity and demonstrating resilience to slight variations in input data, making it an ideal choice for addressing challenging prediction tasks in the NSL-KDD dataset.
[2] By combining convolutional layers and pooling layers, CNN effectively captures spatial structures and local relationships in the NSL-KDD dataset. It autonomously learns and extracts crucial features, enabling efficient classification and discrimination of various attack types. Furthermore, CNN reduces the number of model parameters, thereby decreasing computational complexity. Additionally, it demonstrates a certain level of resilience to slight variations in input data, making it an ideal choice for addressing challenging prediction tasks in the NSL-KDD dataset"

Experimental design

- No comment

Validity of the findings

- No comment

Cite this review as

---

## Round 0.3 · accepted · Accept

We are happy to inform you that your manuscript has been accepted for publication since the reviewers' comments have been addressed.

Reviewer 1 ·

Basic reporting

No comment

Experimental design

No comment

Validity of the findings

No comment

Additional comments

In accordance with the reviewers' comments, the authors have made efforts to implement the necessary adjustments in the study.

Cite this review as

Reviewer 2 ·

Basic reporting

The revision process is completed successfully by authors.

Experimental design

The revision process is completed successfully by authors.

Validity of the findings

The revision process is completed successfully by authors.

Additional comments

The revision process is completed successfully by authors.

Cite this review as